

# Polar tongue of ionisation during geomagnetic superstorm

Dimitry Pokhotelov[1], Isabel Fernandez-Gomez[1], and Claudia Borries[1]

[1]German Aerospace Center (DLR), Institute for Solar-Terrestrial Physics, Neustrelitz, Germany

**Correspondence:** dimitry.pokhotelov@dlr.de

**Abstract.** During the main phase of geomagnetic storms large positive ionospheric plasma density anomalies arise at middle and polar latitudes. A prominent example is the tongue of ionisation (TOI), which extends poleward from the dayside storm-enhanced density (SED) anomaly, often crossing the polar cap and streaming with the plasma convection flow into the nightside ionosphere. A fragmentation of the TOI anomaly contributes to the formation of polar plasma patches partially responsible for the scintillations of satellite positioning signals at high latitudes. To investigate this intense plasma anomaly, numerical simulations of plasma and neutral dynamics during the geomagnetic superstorm of 20 November 2003 are performed using the Thermosphere Ionosphere Electrodynamics Global Circulation Model (TIE-GCM) coupled with the statistical parameterisation of high-latitude plasma convection. The simulation results reproduce the TOI features consistently with observations of total electron content and with the results of ionospheric tomography, published previously by the authors. It is demonstrated that the fast plasma uplift, due to the electric plasma convection expanded to subauroral mid-latitudes, serves as a primary feeding mechanism for the TOI anomaly, while a complex interplay between electrodynamic and neutral wind transports is shown to contribute to the formation of mid-latitude SED anomaly. It is suggested that better representation of the high-latitude plasma convection is needed. The results are discussed in the context of space weather modelling.

## 1 Introduction

In the course of a geomagnetic storm large amounts of solar wind energy and momentum are deposited into the high-latitude ionosphere through the Joule dissipation of magnetosphere/ionosphere currents and auroral particle precipitation (Rodger et al., 2001). During the storm main phase a large positive dayside ionospheric plasma anomaly, known as the storm-enhanced density (SED), arises at sub-auroral mid-latitudes (Mendillo et al., 1972; Buonsanto, 1999; Immel and Mannucci, 2013). The morphology of dayside SEDs have strong seasonal, local time, longitudinal, and other dependencies (Borries et al., 2015). A formation of the SED anomaly is largely (though not exclusively) attributed to the storm-time changes in plasma transport (Prölss, 1995, 2008; Immel and Mannucci, 2013), especially to the uplift of plasma to higher altitudes with longer recombination times. Storm-time changes in plasma/neutral composition and chemistry play greater role in the formation of negative plasma anomalies, which are more common during the storm recovery phase (Rishbeth et al., 1987; Prölss and Werner, 2002).



The key physical mechanism contributing to the storm-time plasma uplifts include: (a) equatorward thermospheric neutral winds driven by the storm-time Joule dissipation (Anderson, 1976; Rishbeth, 1998); and (b) vertical component of the electric $\mathbf{E} \times \mathbf{B}$ plasma convection expanded equatorward to mid-latitudes (Deng and Ridley, 2006; Heelis et al., 2009). Also a horizontal plasma transport due to the poleward expansion of the equatorial plasma anomaly (Tsurutani et al., 2004) or due to the westward plasma drift caused by subauroral polarization streams (Foster et al., 2007) have been invoked to explain the SED anomaly. However, the importance of the last two mechanisms, which involve substantial horizontal plasma transport over mid-latitudes, has been downplayed by Rishbeth et al. (2010) and Fuller-Rowell (2011), respectively, based on considerations of plasma transport times and global plasma density distributions. In this study the vertical uplifts due to (a) neutral winds and (b) expanded $\mathbf{E} \times \mathbf{B}$ convection are considered as the key competing mechanisms for the generation of SED anomalies. We also note that this study is not aiming to explain the formation of SED anomaly.

Focal point of this study is the Tongue Of Ionisation (TOI), which is a storm-time plasma density anomaly originating at the poleward edge of the SED anomaly, spreading anti-sunward across the polar cap, and reaching the nightside auroral zone (Knudsen, 1974; Foster et al., 2005). The TOI anomaly has been observed during large geomagnetic storms using multiple radar systems (Foster et al., 2005) emphasising the role of cross-polar plasma transport by the enhanced $\mathbf{E} \times \mathbf{B}$ plasma convection flow. Using tomographic inversions of Total Electron Content (TEC) observations, the three-dimensional structure of the TOI anomaly has been revealed (Mitchell et al., 2008) and the role of dayside plasma uplift has been demonstrated (Yin et al., 2006). In-situ satellite observations using ion drift instruments during the 20 November 2003 storm (Pokhotelov et al., 2008) suggested that the uplift can be attributed to the equatorward expansion of $\mathbf{E} \times \mathbf{B}$ convection flow. Storm-time observations of cross-polar plasma convection and plasma density using polar cap digital ionosondes (Pokhotelov et al., 2009) and SuperDARN radars (Thomas et al., 2013) demonstrated that sudden changes in the convection regime (e.g., due to rapid changes in the interplanetary magnetic field) can effectively disrupt the formation of TOI anomaly. The fragmentation of TOI anomaly is considered as one of the mechanisms producing polar patches responsible for radio scintillations (e.g., Moen et al., 2013).

Earlier numerical simulations of the SED anomaly demonstrated competing roles of the plasma uplift mechanisms due to neutral winds and electric fields (e.g., Lin et al., 2005; Crowley et al., 2006; Swisdak et al., 2006). Since the mid-latitude SED anomaly provides a source of the uplifted dense plasma for the TOI anomaly, it is reasonable to assume that the same two mechanisms may control the formation of TOI anomaly. However, the SED anomaly covers the entire local day-evening sector and often persists throughout the storm main phase and through an early part of the recovery phase, while the TOI anomaly is relatively narrow in longitude and persists for shorter times during the main phase. With recent developments of higher resolution ionospheric circulation models (e.g., Maute, 2017), it became possible to simulate the dynamics of TOI across the polar cap. (Liu et al., 2016) modelled the development of TOI anomalies during two similar geomagnetic storms of March 2013 and March 2015 using the new release of Thermosphere-Ionosphere Electrodynamics General Circulation Model (TIE-GCM) with $2.5°$ horizontal resolution. Based on the simulations, they concluded that the uplift and transport due to the $\mathbf{E} \times \mathbf{B}$ drifts generally dominates over other possible drivers, such as neutral winds and compositional/chemical changes.

In this study we use an example of the 20 November 2003 geomagnetic superstorm to analyse key mechanisms responsible for the formation and evolution of the TOI anomaly. The 20 November 2003 storm provides an advantage of being an isolated





event driven by a single coronal mass ejection (Zhang et al., 2007). It is among the largest geomagnetic storms observed
by modern space/ground instrumentation, including Global Navigation Satellite Systems (GNSS). Early studies of this storm
using radars (Foster et al., 2005) and GNSS tomography (Pokhotelov et al., 2008) revealed the dynamics and 3-dimensional
morphology of the TOI anomaly. However, self-consistent numerical simulations of the TOI anomaly were not possible at that
time due to resolution limits of the existing ionospheric models and other factors. In this study the high-resolution version of

TIE-GCM model is used to model the TOI anomaly, with the analysis focusing on possible roles of the $\mathbf{E} \times \mathbf{B}$ drifts and neutral
winds. A comparison with earlier GNSS tomography reconstructions is presented, as well as with TEC distributions using
conventional geometric TEC mapping. Limitations of other ionospheric circulation models in reproducing the TOI anomaly
are also discussed, including the models currently used by space weather services.

## 2   Geomagnetic storm of 20 November 2003

In terms of the equatorial ring current disturbance magnitude, the geomagnetic storm of 20-Nov-2003 was the largest storm of
the solar cycle 23 and one of the largest storms recorded by modern instruments, with the disturbance storm time index (Dst)
reaching the value of -422 nT (Zhang et al., 2007). The storm was an isolated event preceded by a ~20-day period of relatively
quiet geomagnetic activity. Following the interplanetary shock arrival at 8:35 UT on 20-Nov-2003, the main phase of the storm
lasts until ~19 UT.

75       During the main phase, the north-south IMF component ($B_z$) turns strongly negative reaching to below -50 nT, while the
dawn-dusk IMF component ($B_y$) increases to +50 nT in the beginning of the main phase, then goes down and turns negative
after 18 UT (Figure 1). With the solar wind speed ($V_{SW}$) exceeding 700 km/s, this IMF configuration should lead to a very
strong two-cell plasma convection pattern. The observed IMF $B_y$ change from positive to negative is expected to alter the
east-west orientation of the "throat" (the entry region) of cross-polar convection channel throughout the main phase (e.g.,

Sojka et al., 1994). During the main phase, the two-cell convection pattern expands dramatically to lower latitudes (to ~ 35°
magnetic latitude), as been also confirmed by in-situ plasma drift measurements using the Defence Meteorological Satellite
Program (DMSP) spacecraft (Pokhotelov et al., 2008). This expanded convection is expected to cause an anomalous vertical
plasma transport at subauroral latitudes due to the resulting vertical component of $\mathbf{E} \times \mathbf{B}$ drift.

## 3   Simulations of the storm

To analyse the ionospheric dynamics during the 20-Nov-2003 storm, simulations have been performed using the Thermosphere-
Ionosphere Electrodynamics General Circulation Model (TIE-GCM) (Richmond et al., 1992; Qian et al., 2014). TIE-GCM is a
first-principle model simulating energy and momentum equations in the coupled thermosphere-ionosphere system. The current
high-resolution version of TIE-GCM, also described as TIEGCM-ICON (Maute, 2017) to support the Ionospheric Connection
Explorer (ICON) satellite, uses the hydrostatic grid with 57 logarithmically spaced pressure levels (1/4 scale height resolution),

covering geopotential heights from ~ 97 km to ~ 600 km, with uniform horizontal 2.5° grid resolution in longitude and latitude.



To facilitate the thermosphere/ionosphere forcing from above and below, TIE-GCM should be coupled with external models. Mean horizontal neutral winds at the lower simulation boundary can be specified according to the Horizontal Wind Model (HWM07; Drob et al., 2008) and atmospheric tides are specified according to the Global Scale Wave Model (GSWM; Hagan and Forbes, 2002, 2003). Most relevant to the high-latitude storm dynamics, the plasma convection pattern is specified accord-

ing to the statistical parameterisations of Heelis et al. (1982) or Weimer (2005). In this study, the Weimer parameterisation (Weimer, 2005) is used with the electrostatic potential expressed as a function of solar wind and IMF parameters measured upstream the Earth's magnetosphere and time-shifted to the bowshock according to King and Papitashvili (2005).

The TIE-GCM simulation is performed throughout the 20-Nov-2003 storm after the 20-day initialisation run to reach the model equilibrium. Following the methodology of Liu et al. (2016), the simulated outputs of 19-Nov-2003 quiet day are

subtracted from the simulated 20-Nov-2003 storm day outputs. The resulting relative $\Delta$TEC[1] anomalies for the 20-Nov-2003 day are shown as a snapshot at 15 UT in Figure 2 and as an animated sequence for the interval 10-23 UT in the Supplements (movie01.avi). For reference, absolute values of TEC are also shown in Figure 2.

The following parameters relevant to storm-time plasma dynamics are also extracted from the TIE-GCM simulation. The height of ionospheric F2 peak (hmF2) and plasma density at the F2 peak (NmF2) are shown in Figure 2. Using the electrostatic

potential ($\phi$), given by the Weimer convection model, horizontal and vertical components of the plasma electric drift ($\mathbf{U}_{E\times B} = -(\nabla\phi \times \mathbf{B})/B^2$) are computed as vector products with the Earth's internal magnetic field ($\mathbf{B}$). Meridional ($V_{E\times B}$) and vertical ($W_{E\times B}$) components of the electric drift are shown in Figure 3 (a snapshot at 15 UT) and as an animated sequence for the interval 10-23 UT in the Supplements (movie02.avi). Meridional neutral winds (V) at pressure levels corresponding to $\sim 120$ km and $\sim 400$ km geopotential heights, and the Joule heating per unit mass (Q Joule), are shown in Figure 4 (a snapshot at 15

UT) and in the Supplements (movie03.avi)

Simulations of the 20-Nov-2003 storm with the Coupled Thermosphere Ionosphere Plasmasphere electrodynamics (CTIPe) model (Fuller-Rowell et al., 1996; Millward et al., 2001) have been also used in this study to compare to the TIE-GCM simulations described above. CTIPe is the first-principle model solving plasma and neutral dynamics on the hydrostatic grid with resolution of $2° \times 18°$ degrees in latitude and longitude, respectively, and 15 pressure levels in the vertical direction going

from the lower boundary at $\sim 80$ km to $\sim 400$ km altitude. The model uses atmospheric forcing given by a simplified version of the Whole Atmosphere Model (Akmaev et al., 2008) and high-latitude electrodynamic forcing specified according to the statistical parameterisations of Weimer (2005). The capability of CTIPe to reproduce SED anomalies during the main phase of 20-Nov-2003 superstorm has been demonstrated in Fernandez-Gomez et al. (2019) for the European sector. In this study, the extended CTIPe run has been used to analyse anomalies in the North American sector, with the results discussed in Section

5.4. The purpose is to see if the operational CTIPe model reproduces similar features as the more computationally demanding research model (TIE-GCM).

---

[1]All the variables starting with $\Delta$ (relative) are obtained by subtracting the quiet day (19-Nov-2009) background.





## 4 Total electron content from GNSS mapping and tomography

The radio signals transmitted by Global Navigation Satellite Systems (GNSS) can be used to retrieve information about iono-spheric plasma anomalies. Due to dispersive properties of the ionospheric plasma, GNSS signals carry information about the
TEC along the signal trajectory. Using thin ionospheric shell approximation (e.g., Horvath and Crozier, 2007) and taking proper care of the receiver and transmitter biases, slant TEC observations by ground GNSS receivers can be converted into the 2D distributions of vertical TEC. GNSS based maps of TEC, using the thin shell transformation, are available from the International GNSS Services (IGS) with typical grid resolutions of $2.5° \times 5°$ in latitude $\times$ longitude (Hernández-Pajares et al., 2009). IGS TEC maps can be directly compared to the results of numerical simulations, though one needs to be careful with arte-
facts caused by sparse/uneven distributions of ground GNSS receivers. Examples of IGS TEC maps during the 20-Nov-2003 storm are presented in Figure 5, also showing a comparison with the distributions of absolute TEC simulated by TIE-GCM. An animated sequence of absolute TEC maps from TIE-GCM simulations and from IGS services for the interval 11-23 UT 20-Nov-2003 is included in the Supplements (movie04.avi).

A tomographic inversion of multiple slant TEC observations is also possible yielding the 3D distribution of plasma den-
sity. The three-dimensional time-dependent algorithm of ionospheric plasma tomography is described by Mitchell and Spencer (2003) and it has been previously applied to reconstruct the high-latitude plasma anomalies during 20-Nov-2003 storm (Pokhotelov et al., 2008), using the network of 60 ground IGS receivers. An additional information about the $\mathbf{E} \times \mathbf{B}$ plasma drifts has been included in the tomographic algorithm uisng the Kalman filters with the Weimer convection model as an a priori information (Spencer and Mitchell, 2007). The distributions of TEC obtained from the tomographic reconstructions are previously pub-
lished and are presented in Figure 6 of Pokhotelov et al. (2008) in the same format and at the same time moments (16 and 18 UT) as TEC distributions from TIE-GCM simulations and from IGS services shown here in Figure 5.

## 5 Discussion

### 5.1 Total electron content

Total electron content maps provide global coverage showing the morphology of plasma anomalies on ionospheric mesoscales
comparable to the horizontal resolution of TIE-GCM simulations presented here ($2.5° \times 2.5°$). However, the TEC mapping experience potential problems at high latitudes due to: (a) sparse/uneven distribution of ground GNSS receivers, (b) singular-ities of the latitude/longitude grid at the geographic poles, and (c) configuration of the GNSS satellite orbits. The network of ground IGS receiver stations available at polar latitudes during the 20-Nov-2003 storm is presented in Figure 2 of Pokhotelov et al. (2008), showing separations between some of the polar cap receivers far greater than the desired horizontal resolutions.
The inclination of GPS satellite orbits of $\sim 55°$ (Samama, 2008) also contributes to the deficiencies of TEC reconstructions in the polar cap region. The tomographic reconstruction algorithm (Mitchell and Spencer, 2003; Spencer and Mitchell, 2007) partially mitigates these deficiencies by using rotated tomographic grids without the polar singularity and by including an a



priory information about plasma convection in the polar cap. Thus the tomographic reconstruction has advantages over the thin shell IGS TEC mapping, providing a more homogeneous solution across the polar cap region.

Taking into account the above limitations, we can compare the simulated TEC distributions with the results of TEC mapping and tomography. As shown in Figure 5 and in the animation (movie04.avi), the TOI anomaly is visible in TEC maps (both from IGS and from TIE-GCM model) starting from $\sim$ 12 UT, though the poleward extension of SED anomaly appears at first some 20° further westward in the TIE-GCM simulations relative to the IGS TEC maps (southern tip of Greenland in the simulations vs east of Iceland in IGS maps). The reasons for this mismatch in the local time / location of the TOI formation are not clear

and will be discussed further in Section 5.4. One has to note that TEC reconstructions are not reliable over the Atlantic ocean sector due to poor GNSS receiver coverage.

The main development of the TOI anomaly (from $\sim$ 13 UT to 18 UT) is seen over the east sector of the United States - Canada, spreading further anti-sunward over the geomagnetic North Pole and northern tip of Greenland. In simulations and in TEC observations the TOI anomaly develops in the same longitudinal sector ($60° - 90°$ W), though the simulated TOI appears

more narrow in longitude and more homogeneous in latitude. After 19 UT the TOI anomaly starts to disintegrate and disappear and the remains of plasma are transported across the polar cap, merging into the nightside auroral TEC enhancement seen over the European sector. Overall, the location and general morphology of the simulated TOI anomaly is remarkably close to the IGS TEC observations and the tomography, except the difference in the TOI onset time/location mentioned earlier. The amplitudes of modelled TEC anomalies (both SED and TOI) appear somewhat higher relative to the observations, confirming

the assessment of Liu et al. (2016) that TIE-GCM generally overestimates the magnitude of positive storm anomalies at high latitudes, though the specific reasons for this overestimation cannot be addressed here.

## 5.2 Plasma uplift dynamics

At first we analyse the dynamics of plasma uplift without looking into specific uplift mechanisms. As most of ionospheric plasma is expected to be confined in the vicinity of F2 peak, it is instructive to compare TEC distributions to the height and

density of F2 peak. The comparison (see Figure 2 and the Supplements) confirms that the F2 peak plasma density (NmF2) largely mimics the behaviour of TEC. In contrast, the change in F2 peak height ($\Delta$hmF2) shows a more complex behaviour. Substantial enhancements of hmF2 (up to 300 km) appear in the following longitude sectors (as referred to 15 UT 20-Nov-2003): (a) central part of the mainland USA west of 80°W, westward of the main SED anomaly; (b) east coast of Canada and towards the geomagnetic North Pole 45-65° W, corresponding to the TOI location; and (c) eastern part of Europe and Central

Asia east of 20° E, in the post-sunset sector. Out of these three major hmF2 enhancements, only the TOI-related enhancement (b) is accompanied by clear increase in plasma density and TEC, while the other two enhancements are accompanied by negative density anomalies. The post-sunset enhancement in hmF2 (c) is considered to be related to a sudden significant increase in hmF2 reported in Borries et al. (2017), which is accompanied by an extreme increase of the equivalent slab thickness. The authors consider intensive plasma transport with strong vertical components at this period of time over the respective region.

The most westward enhancement in hmF2 (a) is due to the early formation of SED anomaly in that sector and does not have clear connection to the TOI anomaly. Some secondary positive/negative anomalies in hmF2 are seen in conjunction/alignment





to the auroral TEC anomalies and will not be discussed here. The main focus here is the clear enhancement of hmF2 coinciding with the positive anomaly in NmF2 and TEC at the poleward edge of SED anomaly and the throat of TOI anomaly, lasting from ∼14 UT to 19 UT.

### 5.3 Electrodynamic vs neutral wind transport

We first focus on the comparison between the modelled relative TEC distributions and the electrodynamic transport parameters shown in Figure 3 and in the Supplements. As indicated by the electric potential distributions, the high-latitude plasma convection pattern greatly expands equatorwards and develops the characteristic two-cell pattern following the southward IMF turn at 11-12 UT. The expanded two-cell convection pattern persists through the storm's main phase reaching the maximum

expansion at 17-18 UT around the minimum of SYMH index. This is consistent with the DMSP satellite observations of $\mathbf{E} \times \mathbf{B}$ convection during this storm (see Figures 5 and 6 in Pokhotelov et al. (2008)). The comparison shows that the Weimer model used in TIE-GCM underestimates the degree of equatorward expansion. Due to IMF $By$ being strongly positive in the early main phase (11-15 UT), the convection "throat" is initially oriented NW-SE, later changing its orientation to NE-SW, when the IMF $B_y$ turns negative around 18 UT. This change in orientation of the convective channel is clearly reflected in the shape of

the TOI TEC anomaly. The influence of east-west convection asymmetry on the TOI anomaly due to the IMF $B_y$ dynamics has been reported before (e.g., Sojka et al., 1994) and it requires further analysis, which is outside the scope of this study. The important feature of electrodynamic plasma transport is the enhancement in vertical electric drift component ($W_{E \times B}$) seen at latitudes from 60°N down to 40-45°N, which accompanies the equtorward expansion of plasma convection. The vertical drift component arises from the $\mathbf{E} \times \mathbf{B}$ convection expanded to latitudes where dipolar magnetic field lines are far from vertical

(e.g., Swisdak et al., 2006). The vertical electric drift maximises in the same longitudinal sector as the TOI anomaly. It maximises at the poleward edge of SED anomaly and in the throat of cross-polar convection channel (∼ 70°W, 50-60 °N), but has a larger E-W extension (∼ 30-100°W) than the TOI anomaly itself. Additionally, enhanced vertical drifts are seen at ∼ 40 °N in a broader range of longitudes extending into the central-western USA sector (west of 90°W). The amplitudes of vertical drifts of ∼ 200 m/s appear to be very large but they are generally consistent with occasional storm-time measurements of large

vertical plasma drifts by the mid-latitude Millstone Hill incoherent scatter radar (Yeh and Foster, 1990; Erickson et al., 2010; Zhang et al., 2017) and the uplifts of F2 peak by ∼ 400 km within 1 hour estimated from tomographic reconstructions during the main phase of 30 October 2003 superstorm (Yin et al., 2006).

During storms, Joule heating in the auroral region changes thermospheric winds and generates so-called storm wind cells (Volland, 1983). The model results show (Figure 4) the enhanced Joule heating near the throat region at about 60 °N, but the

amplitude is small compared to the Joule heating in the night side auroral region (∼ 140°W−120°E). The heating-induced equatorward neutral winds are expected to cause plasma uplift at subauroral latitudes, contributing to the formation of SED (Rishbeth, 1998; Swisdak et al., 2006) and possibly TOI anomalies. The modelled distributions of meridional neutral winds (see Figure 4 and the Supplements) clearly show an enhancements of winds (200-300 m/s at 120 km height and up to 500 m/s at 400 km height) in the longitudinal sector of TOI anomaly, blowing in the anti-sunward (cross-polar) direction, even partially

equatorwards of the heating region (60−80°W). Enhanced equatorward neutral winds are primarily seen in the central-western



USA sector (west of 90°W). We also notice that at the early stage of the TOI formation ($\sim$ 13 UT) the meridional neutral winds are nearly zero at the poleward edge of SED and at the throat of the convective channel, but become polewards later on and appear at higher latitudes. This is an indication that at the poleward edge of SED and in the throat region forcing from the enhanced $\mathbf{E} \times \mathbf{B}$ convection flow is stronger than the forcing from heating-induced winds. The cross-polar neutral wind is

mainly driven by the plasma convection, thus forming the polar cap neutral tongue anomaly (Burns et al., 2004).

### 5.4 Relations to other modelling efforts and space weather applications

After comparing the electrodynamics and neutral wind dynamics, we conclude that the uplift due to the vertical component of enhanced $\mathbf{E} \times \mathbf{B}$ convection is the dominant mechanism forming the TOI anomaly. This is generally consistent with the conclusions of Liu et al. (2016), based on TIE-GCM modelling of two moderate storms driven with the Weimer convection

model. The dominant role of electrodynamic uplift/transport is also confirmed by Huba et al. (2017), who used SAMI3 model driven with the Rice Convection Model (RCM), showing that the realistic TOI anomaly can be reproduced even without including the neutral wind dynamo. The dominant role of electrodynamic plasma uplift in the formation of TOI anomaly does not outrule a complex interplay between electric convection, neutral winds, and other possible mechanisms responsible for the formation of mid-latitude SED anomaly (e.g., Swisdak et al., 2006; Crowley et al., 2006), which is outside the scope of this

study.

The conclusions above are subjected to the right choice of high-latitude $\mathbf{E} \times \mathbf{B}$ plasma convection model. The Weimer parameterisation (Weimer, 2005) used here to drive the TIE-GCM simulations, and also for the earler tomographic reconstructions (Spencer and Mitchell, 2007; Pokhotelov et al., 2008), should provide realistic response to the rapid changes in solar wind / IMF conditions, which could be missing in the case of Heelis parameterisation (Heelis et al., 1982) based on the

3-hour resolution planetary Kp index. Our TIE-GCM simulations repeated for 20-Nov-2003 storm using the Heelis convection parameterisation (not shown here but available on request) demonstrated relatively poor agreement with IGS TEC maps and tomography. Pokhotelov et al. (2008) demonstrated that the statistical Weimer parameterisation may not be able to capture the true extent of equatorward expansion of the $\mathbf{E} \times \mathbf{B}$ convection pattern during the superstorm. The mismatch between the times / longitudes of the early TOI formation (the TOI anomaly appears earlier in time and more eastward in IGS TEC maps

relative to the TIE-GCM simulations, as noted in Section 5.1) is likely due to this underestimation of the $\mathbf{E} \times \mathbf{B}$ expansion. Simulations driven with more realistic convection patterns obtained from, e.g., radar network observations during a specific storm (Wu et al., 2015), or from assimilative models (Lu et al., 2016) may be needed to overcome these deficiencies.

While it is clear that the numerical setup of CTIPe model (namely, the coarse resolution of 18° in longitude) is not ideal for analysing the TOI anomaly, it is beneficial to discuss the results of this model in the context of space weather applications

as the CTIPe is currently used for operational analysis and forecast by the US National Oceanic and Atmospheric Administration Space Weather Prediction Center https://www.swpc.noaa.gov/models (Codrescu et al., 2012). The CTIPe simulation of 20-Nov-2003 storm by Fernandez-Gomez et al. (2019) extended to the North American sector do not show clear TOI developments, though the CTIPe reproduces enhanced neutral wind patterns in the polar cap (Figure A1), similar to those modelled by the TIE-GCM. On the other hand, Pryse et al. (2009) demonstrated that the CTIP model (Millward et al., 1996) was able





to reproduce some features of the TOI anomaly consistent with ionospheric tomography when the simulation was driven by the SuperDARN radar observations of plasma convection. The use of SuperDARN data for driving the simulations was not addressed here but should be exploited in the future.

A fragmentation of the TOI anomaly due to IMF dynamics and other mechanisms has been long attributed to the formation of polar cap plasma patches (Sojka et al., 1994; Carlson Jr. et al., 2004). Climatological studies of ionospheric GNSS scintillations
at high latitudes (e.g., Prikryl et al., 2015) demonstrate strong correlations with the plasma patches, especially near noon in the cusp region and near midnight, i.e, near the exit from cross-polar convection channel[2]. One has to note that polar patches are formally defined as drifting F-region plasma irregularities with horizontal scales $\sim$100 km and densities 2-10 times above the background and could also be formed during geomagnetically quiet times (Moen et al., 2013). Nevertheless, the TOI anomaly is expected to be a dominant source of the high-latitude GNSS disruptions during geomagnetic storms and it needs to be
addressed in space weather applications.

## 6   Summary and conclusions

The feeding mechanisms of the TOI anomaly have been analysed using the simulations of geomagnetic superstorm of 20 November 2003, which have been conducted using the high-resolution version of TIE-GCM ionospheric circulation model with the Weimer parameterisation of high-latitude $\mathbf{E} \times \mathbf{B}$ plasma convection. The simulation results are compared to the IGS
TEC maps and to the results of ionospheric GNSS tomography for this storm event, published earlier by the authors (Pokhotelov et al., 2008). The main conclusions are summarised as following:

(a) The TIE-GCM simulations reproduce the development of polar TOI anomaly consistently with the IGS TEC maps and the tomographic TEC reconstructions. Differences between the model and observations are seen in the early formation of TOI anomaly and in the magnitude/longitudinal extent of TEC anomaly across the polar cap. The results of TIE-GCM simulations
are qualitatively consistent with earlier modelling of less severe geomagnetic storms with TIE-GCM and other ionospheric models (Liu et al., 2016; Huba et al., 2017). The large uplift velocities shown by TIE-GCM near the poleward edge of SED anomaly and in the convection throat agree with earlier ionospheric tomography results and with radar observations of vertical drifts during large storms. The noted differences between the modelled TEC and IGS TEC maps can be attributed to the model deficiencies (especially the $\mathbf{E} \times \mathbf{B}$ convection parameterisations during storms) and to poor GNSS data coverage in the polar
cap. More rigorous data-model comparisons for moderate (more recent) storms with better GNSS coverage is needed.

(b) Simulated distributions of the plasma and neutral dynamics demonstrate that the plasma uplifts of $\sim 200$ m/s due to the high-latitude $\mathbf{E} \times \mathbf{B}$ plasma convection expanded to mid-latitudes appears to be the dominant mechanism responsible for the formation of TOI anomaly. The neutral winds, enhanced during the storm, show the pattern which is not able to actively contribute to the TOI formation. On the contrary, the SED anomaly at mid-latitude is likely to be influenced by both neutral
wind and electrodynamic transport mechanisms.

---

[2]This relates in particular to phase scintillations, with amplitude scintillations having less clear distribution.



(c) Comparisons between TIE-GCM and CTIPe model show that the lower resolution CTIPe model, currently used for space weather operations, is not able to reproduce the TOI anomaly correctly. On the other hand, TIE-GCM simulation of the TOI anomaly also has clear deficiencies. Better model representation of the $\mathbf{E} \times \mathbf{B}$ plasma convection during extreme geomagnetic storms is needed.

*Data availability.* Solar wind data and geomagnetic indices are available from the NASA OMNIWeb portal http://omniweb.gsfc.nasa.gov. IGS total electron content data are available from the NASA CDAWeb portal https://cdaweb.gsfc.nasa.gov/pub/data/gps. TIE-GCM is an open-source model available from the NCAR High Altitude Observatory https://www.hao.ucar.edu/modeling/tgcm. The complete outputs of TIE-GCM simulations for the 20 November 2003 storm performed at DLR are available upon request to the corresponding author.





*Author contributions.* DP performed TIE-GCM simulations and compiled the manuscript. IFG performed CTIPe simulations and analysed IGS TEC data. CB provided an expertise on mid-latitude ionospheric storm response and directed the study.

*Competing interests.* The authors declare that there are no competing interests.

*Acknowledgements.* The authors are grateful to Philip Erickson from MIT Haystack Observatory for providing insights into storm-time observations of vertical plasma drifts by the Millstone Hill incoherent scatter radar.



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

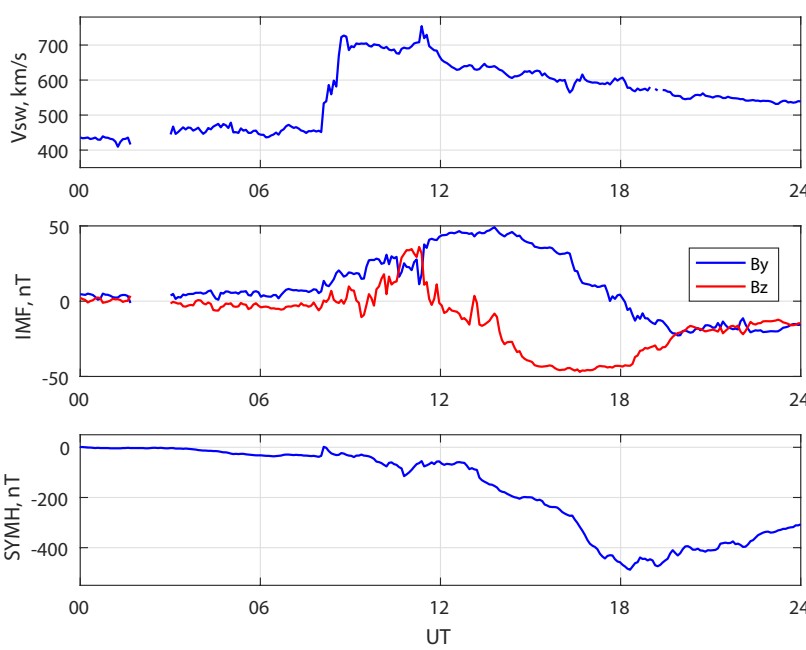

**Figure 1.** Solar wind speed (top), interplanetary magnetic field components (middle), and symmetric horizontal component disturbance index (bottom) during the 20 November 2003 storm.



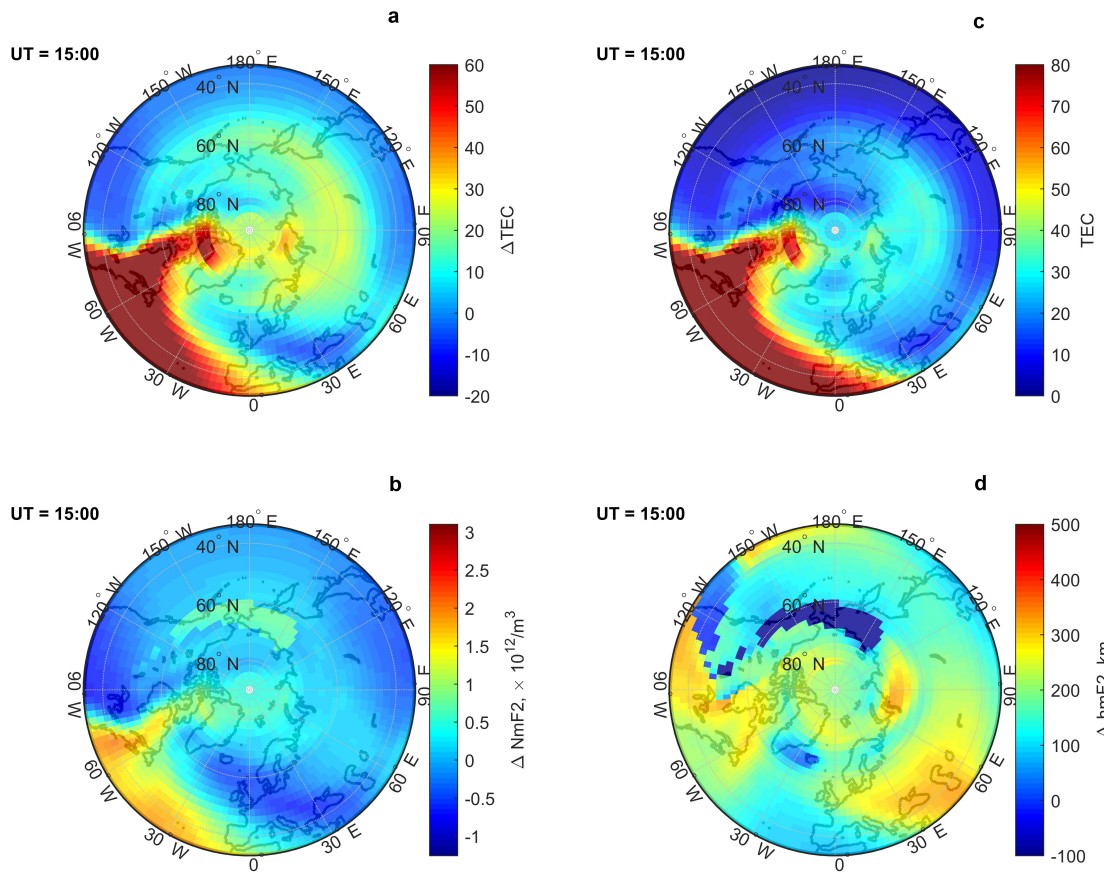

**Figure 2.** Modelled TIE-GCM distributions of relative $\Delta$TEC (a), plasma density at the F2 peak (b), absolute TEC (c) and the height of F2 peak (d) at 15 UT 20-Nov-2003.

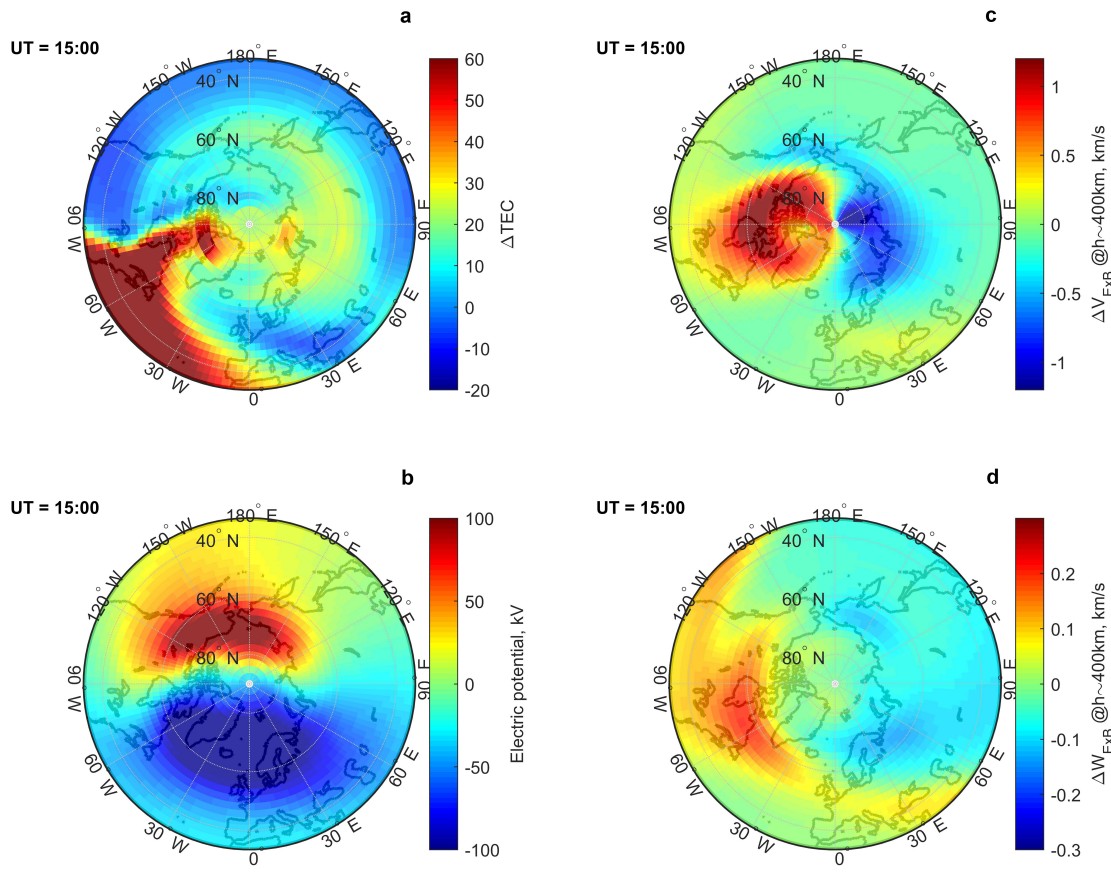

**Figure 3.** Modelled TIE-GCM distributions of relative $\Delta$TEC (a), electrostatic potential (b), relative horizontal (c) and vertical (d) components of the $\mathbf{E} \times \mathbf{B}$ convection flow at 15 UT 20-Nov-2003.

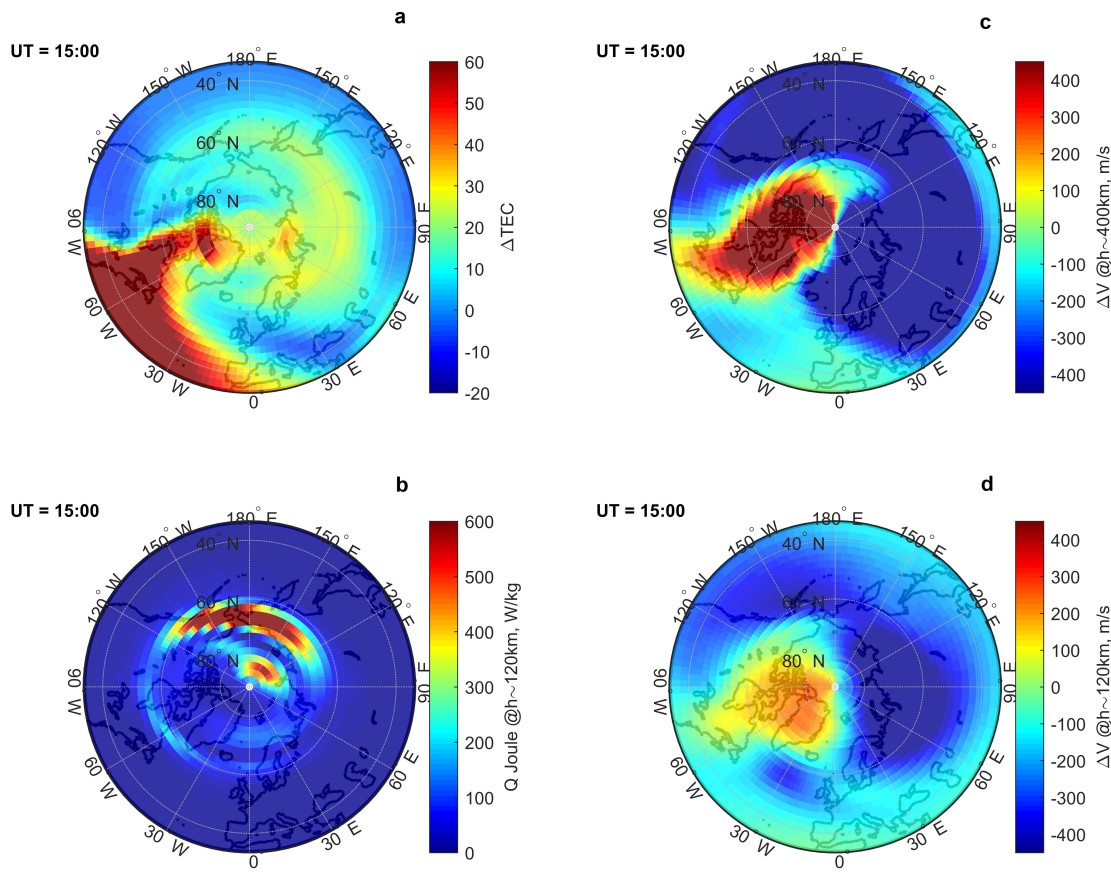

**Figure 4.** Modelled TIE-GCM distributions of relative ΔTEC (a), Joule heating, and relative meridional neutral winds different levels (c)-(d) at 15 UT 20-Nov-2003.

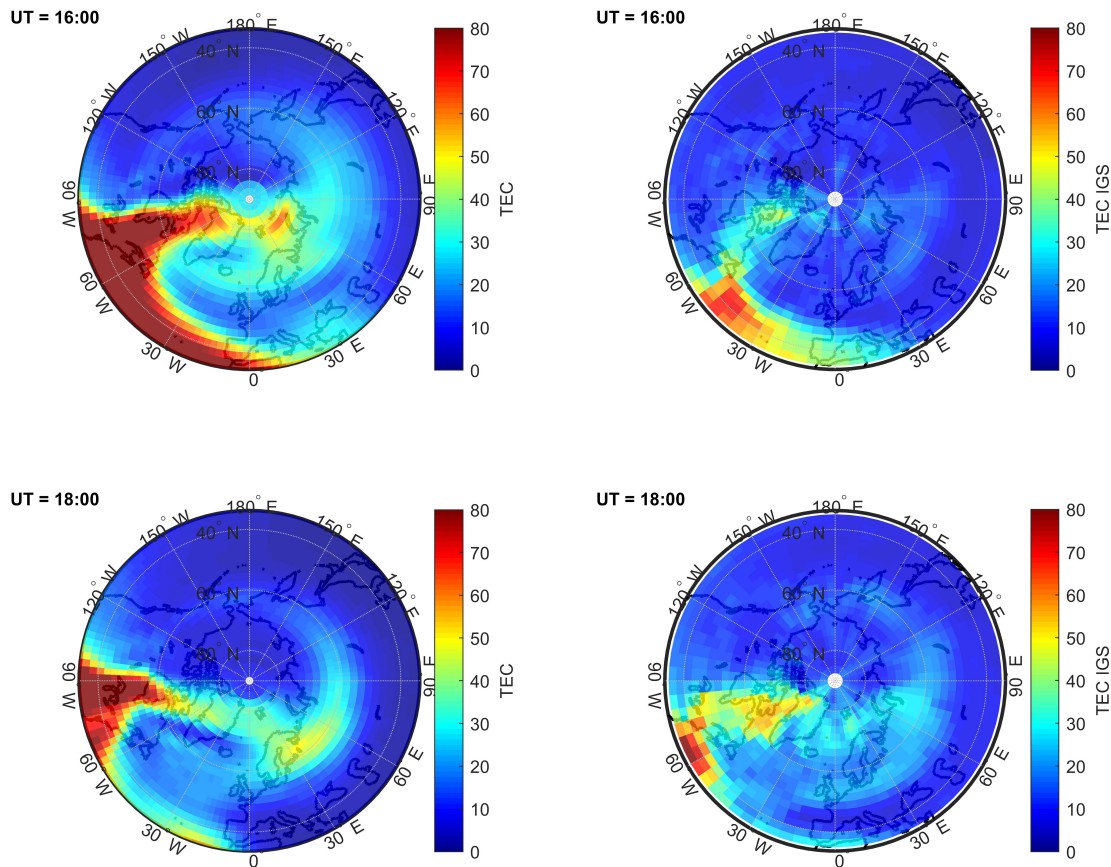

**Figure 5.** TEC distributions obtained from TIE-GCM simulations (left column) and from IGS services (right column) at 16 UT (top raw) and 18 UT (bottom raw) during 20-Nov-2003 storm.



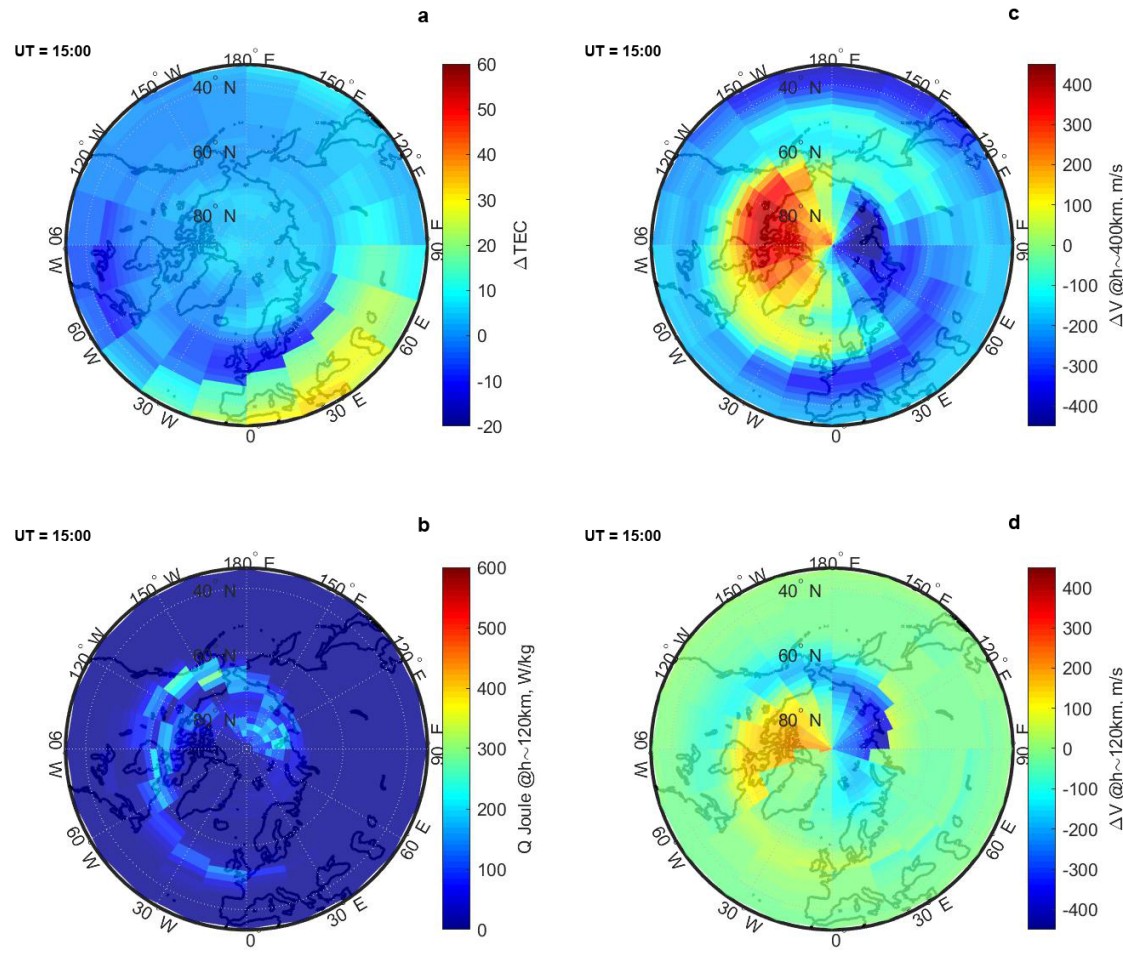

**Figure A1.** Modelled distributions of relative ΔTEC (a), Joule heating (b), and relative meridional neutral winds at different levels (c)-(d) at 15 UT 20-Nov-2003 obtained from the CTIPe simulations.