# Peer review of "Polar tongue of ionisation during geomagnetic superstorm"

_Annales Geophysicae, 2021_

## Author Response (AR1)

**Reply to the Anonymous Referee #1**

We appreciate the referee comments. Below is the detailed response, with the referee's original comments included in *italic* typeface.

*While the topic itself is interesting, I don't think that the research objective and conclusion of this manuscript are new. The main objective of this work is to investigate the role of ExB drift and wind for the vertical uplift of SED. However, a number of recent simulation works have already addressed this topic (Liu et al., 2016, 2017; Dang et al., 2019, Klimenko et al., 2019, Jiang et al., 2020). Those works have shown that the upward component of the ExB drift plays a primary role in formation of the SED and TOI. These works have also discussed the role of wind. I don't think that the current manuscript clearly presents significantly new finding on SED or TOI. The authors should define a unique research objective and establish results that do not overlap the existing works.*

The novelty is in the modelling of TOI formation during the extreme (superstorm) event. We agree that the key mechanisms responsible for the plasma uplift, i.e. the vertical component of the ExB drift and the equatorward neutral winds, have been identified in earlier studies. Some numerical studies have been done already in early 2000s to illustrate the interplay between the two mechanisms. These early works are already discussed and referenced in the Introduction. The recent simulation study by Liu et al. 2016 was already referenced in the article and discussed. We have now included the refence to Klimenko et al., 2019. It is highly relevant and we discussed the implications. Dang et al., 2019 work is focused on the splitting of the TOI ("double tongues") that can't be addressed in our work (Dang et al. used experimental setup of TIE-GCM with very high horizontal resolutions of 0.6 x 0.6 deg). We referenced it as well. Jiang et al. 2020 is focused on the mid-latitude SED anomaly, that is only remotely relevant to the topic. All of the mentioned modelling studies simulated relatively moderate geomagnetic storms. The storm of March 2015 (the largest in the 24th solar cycle), modelled by Liu et al. 2016 and Klimenko et al. 2019, had Dst minimum of -226 nT. Such storms are not in the category of "great storms", commonly defined as Dst below ~ -300 nT (Kamide et al., 1997). To our knowledge, the current study is the only attempt so far to model the TOI formation with a physics-based ionospheric model during a great storm (superstorm) event. The magnetosphere-ionosphere interactions in general, and the formation of SED/TOI in particular, are expected to be quantitatively and qualitatively different during great storms (e.g., Kamide et al., 1997; Yin et al., 2006; Pokhotelov et al., 2008). This defines the significance of our study, both scientifically and in the context of space weather modelling. We demonstrated that the ExB uplift is the dominant mechanism during great storms, at least during this particular superstorm (minimum of -422 nT), which is the largest storm recorded by modern instrumentation. In contrast, for moderate storm simulations, neutral wind effects in the polar cap could be noticeable (e.g., Klimenko et al., 2019). We further clarified the novelty of our study in the text.

*The simulated TEC does not decay with latitude but has a local peak near noon at 80 degrees latitude at 15 UT. Please discuss the cause of this peak. Is this because of cusp precipitation? How much does precipitation impact the formation of the TOI?*

The local peak at 80 degrees noted by the referee is not a permanent feature. In Figure 1 below we show a sequence of 20 min snapshots before and after 15 UT. The noted peak is probably caused by a non-stationarity of the TOI formation/transport, e.g., due to IMF fluctuations, leading to a local plasma enhancement. It is unlikely to be caused by cusp precipitation, that is to our knowledge not

specifically implemented in the TIE-GCM simulation. There are also possible plasma transport problems due to a grid singularity at the pole. We included further discussion in the text.

[Figure]

Figure 1: A sequence of modelled TEC from 14:00 UT to 15:40 UT.

*The authors discussed the local time difference of the simulated and observed SED/TOI. However, the magnitude difference is not discussed. The simulated SED/TOI has much higher TECU and spreads much wider local time. Also the simulated TOI extends to the nightside but the observed TOI disappear near the pole. Please discuss why the simulation overestimated the SED/TOI.*

The overestimation of TEC magnitudes by TIE-GCM is already discussed (lines 170-172). The same overestimation of high-latitude positive anomalies is reported by Liu et al. (2016) for moderate storms. The exact reasons for such overestimation are currently unknown and should be addressed in a model validation work. The observed TOI in IGS TEC maps is poorly represented past the north pole due to the lack of ground receivers in the Arctic ocean area, thus we could not definitely compare IGS TEC maps with the simulations in this region. We commented on this in the text.

*The simulation runs used statistical input parameters. While this approach is reasonable, the manuscript only compares to the observed TEC and does not evaluate errors of other parameters. It is desired to incorporate convection and precipitating particle observations (such as SuperDARN, DMSP and POES) and discuss errors of the simulation.*

We agree with the referee that constraining the simulations by real observations of plasma convection from SuperDARN and satellites could be a fruitful approach. We are planning to do this in the future. Previously, the SuperDARN and DMSP convection data were analysed during the Nov. 2003 superstorm (Foster et al., 2005; Pokhotelov et al., 2008). However, the SuperDARN coverage in 2003 was not optimal and also the DMSP SSIES instruments (ion dynamics) experience problems during great storms (Pokhotelov et al., 2008). We would prefer to do such simulations in the future starting with moderate, as well as more recent, storm events. Adding precipitating particle observations from DMSP or POES should be relatively less important, as the uncertainties in specifying the convection electric fields are generally greater (e.g., Pedatella et al., 2018).

**References**

Foster et al., Multiradar observations of the polar tongue of ionization, Journal of Geophysical Research: Space Physics, 110, https://doi.org/10.1029/2004JA010928, 2005.

Kamide et al., Magnetic Storms: Current Understanding and Outstanding Questions. In Magnetic Storms, AGU Monograph (eds. B.T. Tsurutani, W.D. Gonzalez, Y. Kamide and J.K. Arballo), https://doi.org/10.1029/GM098p0001, 1997.

Pokhotelov, D., Mitchell, C. N., Spencer, P. S. J., Hairston, M. R., and Heelis, R. A.: Ionospheric storm time dynamics as seen by GPS tomography and in situ spacecraft observations, Journal of Geophysical Research: Space Physics, 113, https://doi.org/10.1029/2008JA013109, 2008.

Pedatella, N., Lu, G. and Richmond, A. D., Effects of high-latitude forcing uncertainty on the low-latitude and midlatitude ionosphere. Journal of Geophysical Research: Space Physics, 123, 862–882. https://doi.org/10.1002/2017JA024683, 2018.

Yin, P., Mitchell, C., and Bust, G.: Observations of the F region height redistribution in the storm-time ionosphere over Europe and the USA using GPS imaging, Geophysical Research Letters, 33, https://doi.org/10.1029/2006GL027125, 2006.

**Reply to the Anonymous Referee #2**

We appreciate the referee comments. Below is the detailed response, with the referee's original comments included in *italic* typeface.

*line 88: The authors use TIEGCM V2.0 which supports the 2.5deg resolution and ¼ scale height vertical resolution. TIEGCM-ICON is based on V2.0 but offers additional lower boundary forcing options.*

Correct, we use TIEGCM V2.0. We clarified this in the text.

*Line 105: I assume that the ExB drift components are expressed in geographic directions but it is not totally clear. Maybe it would be good to explicitly state it.*

ExB drift components are expressed in geographic coordinates. We now stated this explicitly in the text.

*Line 115: "given by a simplified version of the Whole Atmosphere Model" The word "simplified" does not add any value and I suggest to state what wave forcing is included to be more specific.*

We have provided more specific description of the atmospheric forcing. "The atmospheric forcing is specified according to the Whole Atmosphere Model (WAM) (Akmaev et al., 2008). WAM fields (neutral temperature, zonal and meridional neutral winds) are averaged in every local hour sector of a given month and thus contain the monthly-averaged mean winds and tides." We also added an acknowledgment to Mariangel Fedrizzi and Mihail Codrescu from the NOAA Space Weather Prediction Center, who consulted us on the issue.

*Line 120: Please double check if CTIPe is still operational or if WAM/IPE is the follow on.*

We confirmed that CTIPe is still used for the NOAA SWPC operations as one of the tools for the total electron content forecast:

https://www.swpc.noaa.gov/products/ctipe-total-electron-content-forecast

WAM-IPE is currently used as the experimental test product:

https://www.swpc.noaa.gov/products/wam-ipe

*Line 120: "computationally demanding" Why do the authors add this, because of the resolution? I would assume that the two models are similar- maybe CTIPe which includes a plasmasphere takes longer, but TIEGCM does not have this physics included.*

The operational CTIPe model used in this study has resolution of 2 deg x 18 deg. On a single core it computes 24 hrs in ~10 min, which is substantially faster than TIEGCM V2.0 with 2.5 x 2.5 deg resolution used here. We agree that for the same resolution the two models should have comparable performance. The purpose here was to compare the features of TOI reproduced by the research run of TIEGCM with those produced by the operational run of CTIPe. We modified the text, to avoid the impression that we are comparing computational performance of the two models.

*Line 138 uisng -> using*  Corrected.

---

## Author Response (AR2)

**Second reply to the Anonymous Referee #1**

We appreciate the referee comments. Below are the detailed responses, with the referee's original comments included in *italic* typeface.

*The authors have clarified that the main difference from the existing works is the investigation of the superstorm. The usage of the superstorm is indeed mentioned throughout the paper, but the abstract and conclusion do not clearly state the uniqueness of the study. The abstract and conclusion should stress that the past findings are from moderate storms and did not investigate superstorms. The current abstract and conclusion only describe the findings that were also seen in the moderate storms. The abstract and conclusion should also clearly describe what results in the superstorm simulation are different from the moderate storms. The only brief discussion about the differences between the superstorm and moderate storm simulations is made at line 251-254. This discussion should be expanded by detailing differences in TEC, convection and wind between the superstorm and moderate storm. Neutral wind, composition and TEC suppression are briefly mentioned, but the differences from the moderate storms should be stated quantitatively.*

The abstract, as well as the discussion and conclusions, have been expanded accordingly to emphasise the results specific to the Nov. 20 superstorm, and to compare with earlier simulations of smaller storms. Since TIEGCM simulations of relatively smaller storms were already published (in particular, Liu et al., 2016, referenced in the paper), we refer to the published results for comparisons. However, it is beyond the scope of this study to provide a detailed quantitative analysis of the differences in ionospheric responses to superstorms vs moderate storms. We believe such quantitative analysis is not possible with the current state of modelling and observations.

*One of the reasons why the existing papers of TOI did not simulate superstorms is that it is difficult to know reliable convection pattern in extreme magnetic conditions. The convection model in TIEGCM is not calibrated against superstorms. As the authors mentioned, convection data such as SuperDARN is sparce in superstorms. How reliable is the convection model in TIEGCM in superstorms? How much do errors in convection influence on the results?*

As already emphasised in the current paper, the Weimer convection model in TIEGCM underestimates the degree of equatorward expansion of the convection pattern. The actual degree of the expansion was assessed in our early study of Nov. 2003 storm using DMSP satellite passes (Pokhotelov et al., 2008, referenced in the paper). Unfortunately, no suitable instruments operated in 2003 to allow a construction of better global convection models. Since that, the SuperDARN radar network has been expanded to lower latitudes, but no new superstorms have happened. Such an error estimate for the current superstorm of Nov. 2003 would be highly speculative, mainly because the convection data are poor. We will address the topic in future works using the expanded SuperDARN convection data from relatively smaller storms.

*Dst = -226 nT storm is not a moderate storm. In Gonzalez et al. [1994]'s widely used definition, Dst < -100 nT storms are intense storms. "intense" storms should be used instead of "moderate" storms. https://doi.org/10.1029/93JA02867*

The referee points out correctly that all the storms with Dst < -100 nT are commonly classified as "intense", while the storms with -50 nT < Dst < -100 nT are classified as "moderate" (e.g., Tsurutani and Gonzalez, AGU Monograph 98, 1997, https://doi.org/10.1029/GM098p0077). Accordingly, the

storm of March 2015 with the Dst minimum of -226 nT should be classified as "intense". However, we described the storm as "relatively moderate" in comparison to the much greater superstorm of Nov. 2003. We have now corrected the use of "moderate" throughout the text, by either removing the term or making it clear that we mean a "relatively moderate" event in comparison to great/superstorm events.